# CVAPPS: A Cross-Sectional Study of SARS-CoV-2 Vaccine Acceptance, Perceptions, and Post-Vaccination Side Effects among Rheumatic Disease Patients in Kuwait

**DOI:** 10.3390/vaccines11030666

**Published:** 2023-03-15

**Authors:** Fatemah Baroun, Anwar Albasri, Fatemah Abutiban, Heba Alhajeri

**Affiliations:** 1Rheumatology Unit, Al-Jahra Hospital, AlJahra, Kuwait; 2Rheumatology Unit, Al-Sabah Hospital, Alsabah, Kuwait; albasri.a.q@hotmail.com; 3Rheumatology Unit, Jaber Alahmad Hospital, South Surra, Kuwait; fmaabutiban@moh.gov.kw; 4Rheumatology Unit, Mubarak Al-Kabeer Hospital, Jabriya, Kuwait; hebah.alhajeri@gmail.com

**Keywords:** rheumatic disease, SARS-CoV-2 vaccine, post-vaccination, safety, Kuwait

## Abstract

**Purpose:** We analyzed data collected for ascertaining severe acute respiratory syndrome coronavirus 2 (SARS-CoV-2) vaccine acceptance, perceptions, and post-vaccination side effects among Kuwaiti inflammatory rheumatic disease (IRD) patients. **Methods:** The current cross-sectional study was conducted on patients attending governmental rheumatology clinics across seven hospitals in Kuwait between July and September 2021. We included adults of both sexes who are national/residents of the state of Kuwait and who had a confirmed diagnosis of any IRD disease. Data on patients’ demographics, history of IRD, SARS-CoV-2 infection, vaccination status, as well as post-vaccination side effects and disease flare were collected from the included participants through a self-administered questionnaire. Stata MP/17 for macOS was used for statistical analyses. **Results:** We included 501 IRD patients, a group with a mean age of 43.38 years and a mean disease duration of 10.46 years. The majority of the included patients were female (79.8%), and the most common primary rheumatology diagnosis was rheumatoid arthritis (42.5%), followed by spondyloarthritis (19.4%) and systemic lupus erythematosus (19.0%). One hundred and five patients (21.0%) had SARS-CoV-2 infection confirmed by PCR-positive swab; of them, 17 patients were hospitalized. None of the included patients were using steroids alone. cDMARDs, bDMARDs, and sDMARDs were reported in 37.3%, 18.0%, and 3.8% of the patients, respectively. Three hundred and fifty-one patients (70.1%) were vaccinated; 40.9% received Pfizer/BioNTech, whereas 28.7% received AstraZeneca/Oxford vaccines. Fear that the vaccination will aggravate the condition or interfere with the present therapy and concerns about its effectiveness as well as its side effects were the most prevalent causes for refusing to accept the SARS-CoV-2 vaccine. Other patients were worried about the paucity of the data because individuals with IRD had been omitted from earlier research, resulting in a dearth of information. The commonly reported post-vaccination side effects were body ache/pain, fatigue, and pain at the injection site (32.1%, 30.3%, and 29.7%, respectively). IRD flare post-SARS-CoV-2 vaccination was self-reported in only 9 patients, and 342 did not report a flare. **Conclusions:** This study’s findings highlight that SARS-CoV-2 vaccines have an acceptable safety profile, with the majority of their side effects being temporary and mild. The occurrence of flares was low after immunization. Reassurance and trust in the safety of the SARS-CoV-2 vaccination in IRD patients should be reassuring to rheumatologists and vaccine recipients.

## 1. Introduction

The coronavirus disease 2019 (COVID-19) pandemic, caused by a novel coronavirus identified as severe acute respiratory syndrome coronavirus 2 (SARS-CoV-2), has wreaked havoc on healthcare systems since its start in December 2019, and it continues to pose a significant concern [1,2,3]. The severity of the illness may be reduced if efficient and safe vaccines are widely used [4]. COVID-19 seems to be associated with an increased risk of severe disease and death in inflammatory rheumatic disease (IRD) patients with rheumatoid arthritis (RA), systemic lupus erythematosus (SLE), systemic sclerosis (SSc), and idiopathic inflammatory myositis when compared to healthy people [5,6,7]. It was reported that age and comorbidities were associated with severe COVID-19 disease in the IRD cohort; nevertheless, the impact of biological treatments on COVID-19 outcomes is unknown [8].

Many trials show that some immunosuppressants or biologic drugs may reduce the immunogenicity of vaccinations, even though the SARS-CoV-2 vaccines’ safety profile in IRD patients has not been well-investigated [9,10,11,12,13,14]. IRD patients infected with SARS-CoV-2 and on immunosuppressants have a higher risk of developing severe disease and needing additional hospitalization periods [15]. microRNA (mRNA) as well as recombinant adenovirus formats are revolutionary vaccination methods. Nevertheless, in clinical studies, exclusively healthy or immunocompetent individuals were evaluated for SARS-CoV-2 vaccination immunogenicity and safety. To date, only a few phase IV studies have been conducted to assess the safety and immunogenicity of vaccinations for IRD patients.

In this study, we aimed to estimate the relative numbers of IRD patients who received the SARS-CoV-2 vaccine in the State of Kuwait. Furthermore, this study highlights SARS-CoV-2 vaccine acceptance, perceptions, and post-vaccination side effects among IRD patients. This data should help healthcare policymakers, public health practitioners, physicians, and rheumatologists increase vaccination uptake and awareness among IRD patients.

## 2. Methods

We reported the current manuscript in accordance with the Strengthening the Reporting of Observational Studies in Epidemiology (STROBE) statement guidelines for cross-sectional studies [16]. Ethical approval for this study was provided by the Ministry of Health (MOH) of the Kuwait ethics committee (IRB number: 1662/2021). The study was conducted in accordance with the ethical principles of the Declaration of Helsinki, and all patients provided informed consent [17].

### 2.1. Study Population and Data Source

This cross-sectional study targeted patients with IRD in the State of Kuwait. Patients who attended rheumatology clinics and daycare units across seven governmental hospitals in Kuwait from 12 July 2021 to 4 September 2021 were invited to answer a web-based questionnaire. Adult patients aged ≥21 years of both sexes who were nationals and residents of the state of Kuwait and with a confirmed diagnosis of any IRD disease were eligible to be included in the current study.

### 2.2. Data Sources/Measurement

The patients’ data were collected through a self-administered questionnaire (Appendix A). The questionnaire included demographic characteristics (including age, gender, smoking, education level, employment status, and marital status), rheumatological disease data (type of disease, disease duration, and activity), COVID-19-related data (SARS-CoV-2 infection and outcomes), vaccination status (prior influenza or pneumococcal vaccinations as well as SARS-CoV-2 vaccine acceptance, source of information, and opposition to receiving the vaccine), and post-vaccination side effects and disease flare. The questionnaire was modified from its original form, and its validity was confirmed in pilot testing.

### 2.3. Study Size

Because the primary endpoint of this study was to determine the rates of vaccination status and post-vaccination side effects and flares, we calculated the sample size based on the recommended methods for cross-sectional studies [18]. To detect proportion rates of 50% in an undetermined population size with 5% precision, the minimum sample size required for this study was 398 patients.

### 2.4. Statistical Analysis

Qualitative data such as sex, vaccination status, post-vaccination side effects, and post-vaccination disease flare are expressed as frequencies and percentages. Quantitative variables such as age and disease duration are expressed as the mean and standard deviations if normally distributed, or as median and interquartile ranges if not normally distributed. The independent t-test and Mann–Whitney test were used to compare continuous variables as appropriate, whereas Pearson’s Chi-squared test was used to compare categorical variables. Analysis was done using Stata MP/version 17 for macOS. All tests were two-sided. Statistical significance was set at a *p*-value < 0.5.

## 3. Results

### 3.1. Demographic Characteristics of the Included Participants

During the study period, 547 participants consented to participate in the survey; 501 met the inclusion criteria and were included in the current study (Figure 1).

Their ages ranged between 22 and 78 years, with a mean of 43.38 years (SD = 11.15). Most included participants were female (n = 400, 79.8%) and had higher educational levels (n = 264, 52.7%). Of the participants, 149 (29.7%) were healthcare providers. The most common primary rheumatology diagnosis among the included patients was RA in 213 patients (42.5%), followed by spondyloarthritis (SpA) in 97 patients (19.4%) and SLE in 95 patients (19.0%). The mean disease duration was 10.46 years (SD = 8.54). Regarding medications, none of the included patients were using steroids alone, whereas 187 patients (37.3%), 90 patients (18.0%), and 19 patients (3.8%) used cDMARDs, bDMARDs, and sDMARDs, respectively. Seven patients (1.4%) were on B-cell-depleting therapy, as shown in Table 1. Positive PCR was self-reported in 105 patients (21.0%). Following SARS-CoV-2 infection, 17 patients were hospitalized, and 86 were not.

### 3.2. SARS-CoV-2 Vaccine Status among the Included Participants

Of the included participants, 351 patients (70.1%) were vaccinated. Of them, 205 (40.9%) had received Pfizer/BioNTech, whereas 144 (28.7%) had received AstraZeneca/Oxford vaccines. On the other hand, 133 (29.9%) patients reported not being vaccinated. Reasons for refusing to accept the SARS-CoV-2 vaccine are shown in Figure 2.

Concerns about vaccine side effects, disease flare, and treatment interaction were the most reported reasons for vaccine refusal. Other patients were concerned about the scarcity of data, which was likely because patients with IRD had been excluded from previous vaccine trials.

### 3.3. Vaccination Status Based on Demographic Characteristics

Participants’ demographic characteristics based on their vaccination status are shown in Table 2. We observed no statistical differences between both groups regarding age, gender, and disease duration (all *p* > 0.05). There was a significant difference between patients with higher education levels regarding vaccine status (203 patients were vaccinated vs. 57 who were not, *p* = 0.003). In total, 91% of the patients refused previous SARS-CoV-2 vaccines.

SARS-CoV-2 vaccination was reported in 154 (43.9%), 69 (19.7%), and 57 (16.2%) patients with RA, spondyloarthritis (SpA), and SLE, respectively, compared to 53 (39.6%), 28 (20.9%), and 33 (24.6%) who were not, respectively, with a *p*-value of 0.042. The rate of vaccination was higher in patients who consulted their rheumatologist regarding the vaccine (*p* < 0.001) and/or read the Kuwait Association of Rheumatology (KAR) online information (*p* = 0.005). The rate of SARS-CoV-2 vaccination was higher in participants who received prior influenza vaccines (*p* < 0.001). Moreover, concerns about getting a SARS-CoV-2 infection were reported more in the vaccinated group than in the non-vaccinated group (*p* = 0.039).

### 3.4. Side Effects

The most commonly reported post-vaccination side effects were body ache/pain (n = 161, 32.1%), fatigue (n = 152, 30.3%), and pain at the injection site (n = 149, 29.7%). Side effects lasted fewer than five days in the majority of participants (202/501, 40.3%). The frequencies of post-vaccination side effects are shown in Figure 3. No complications were reported in 59 of the vaccinated patients (16.8%).

### 3.5. Flares Post-SARS-CoV-2 Vaccination

IRD flares post-SARS-CoV-2 vaccination were reported in only 9 patients, and 342 did not report a flare. There were no statistically significant differences between both groups in terms of age (*p* = 0.861), type of vaccine (*p* = 0.837), gender (*p* = 0.1), and type of rheumatological diseases (*p* = 0.648), as shown in Table 3.

## 4. Discussion

Comorbidities are strong predictors of severe SARS-CoV-2 infection and of adverse clinical outcomes. Previous reports show that patients with diabetes, hypertension, cardiovascular diseases, and respiratory illnesses have a higher risk of hospitalization, intensive care unit (ICU) admission, mechanical ventilation, and mortality [19]. Likewise, IRD was linked to a higher risk of severe SARS-CoV-2 infection; recent data highlight that disease activity is an independent predictor of COVID-19-related hospitalization and mortality [20]. Moreover, in the IRD cohort, immunosuppressive drugs increased the risk of COVID-19-related hospitalization and mortality [21]. Mass SARS-CoV-2 vaccination is the most effective modality to mitigate the risk and consequences of the pandemic in the general population. As a result of the rapid production of tolerated and efficacious SARS-CoV-2 vaccines, major issues concerning vaccination efficacy and safety in patients with IRD, particularly those who are using immunomodulation, were raised. SARS-CoV-2 vaccination studies initially excluded IRD patients and those on immunomodulation, leaving physicians as well as patients with concerns [22]. These concerns were related mainly to the interaction between the underlying immune mechanisms of IRD or of immunosuppressive treatment and vaccines [23]. However, later clinical data confirmed the safety of mRNA and inactivated SARS-CoV-2 vaccines in patients with IRD [9,24,25]. Thus, clinical guidelines recommend SARS-CoV-2 vaccines for patients with IRD, with certain adjustments in therapeutic regimens [26]. Still, a lower immunogenic response to the SARS-CoV-2 vaccine is concerning in IRD patients. In a previous study of 133 patients with autoimmune diseases (28.6% had rheumatoid arthritis, 15% had spondyloarthritis, and 11% had systemic lupus erythematosus), many patients had a strong antibody response to the two doses of mRNA vaccinations [27]. However, it was three times lower than the responses seen in healthy controls. A recent meta-analysis of 47 trials showed that SARS-CoV-2 vaccination humoral and cellular immunogenicity decreased in IRD patients but significantly improved following the second vaccine dosage. On the other hand, anti-CD20 treatment was linked to decreased humoral immunogenicity [4]. Consequently, the American College of Rheumatology (ACR) recommended a fourth dose of mRNA vaccines for IRD patients.

It is imperative, in return, to ensure that IRD patients have a high level of acceptance towards SARS-CoV-2 vaccines and to investigate the possible predictors of low acceptance levels. In this study, we assessed SARS-CoV-2 vaccine acceptance, perceptions, and post-vaccination side effects among IRD patients. This data should assist healthcare policymakers, public health practitioners, physicians, and rheumatologists in increasing vaccination uptake and awareness in IRD patients. This survey-based study received a total of 501 eligible responses, with the majority being women. Previous survey-based research on accepting the SARS-CoV-2 vaccines among IRD patients found similar results [23,28,29], which may be attributed to the predominance of women in the IRD cohort.

In the present survey, we found that a high percentage of IRD patients (70.1%) were vaccinated, mainly with Pfizer/BioNTech and AstraZeneca/Oxford vaccines. In line with our findings, a survey-based study recruited IRD patients enrolled in a prospective cohort study in the Netherlands to assess the COVID-19 impact on the IRD population. The results show that 62% of the IRD patients were willing to get vaccinated or had already been vaccinated in December 2020, which increased to 91% in April 2021 and 95% in August 2021 [23]. Another survey from Australia found that 65% of IRD patients were willing to be vaccinated [29]. A slightly lower percentage (52%) was reported in India [28], whereas in a previous large cross-sectional study in Arab countries, including Kuwait, 29% of IRD patients were vaccinated [30].

The acceptance and reluctance rates of SARS-CoV-2 vaccinations differ significantly across people and countries. An analysis of low- and middle-income countries revealed an acceptance rate of 80.3%, which is higher compared to 64.6% in the US and 30.4% in Russia [31]. A group of 344 individuals with rheumatic and musculoskeletal diseases in Italy had a 54.9% acceptance rate for SARS-CoV-2 immunization [32]. This study showed that vaccination rates among IRD patients in Kuwait are acceptable (70.1%). Felten et al. [33] documented that 686 out of the 1266 patients with systemic autoimmunity or IRD (54.2%) accepted vaccination against SARS-CoV-2, 32.2% were uncertain, and 13.6% refused to get vaccinated.

Boekel et al. highlighted in their study that two factors were associated with a higher acceptance rate of the vaccination: advice from rheumatologists and prior pneumococcal vaccinations or vaccinations against influenza [23]. Our study found that high education, consulting rheumatologists regarding the vaccine, reading trusted online information, and prior influenza vaccines were positively associated with increased vaccine acceptance.

Felten et al. [33] showed that patients with systemic autoimmunity or IRD expressed a desire to be immunized against SARS-CoV-2 to protect themselves, their families, and the public, with percentages of 67.1%, 54.2%, and 62.5%, respectively. Women were rather less inclined to be vaccinated than men (71.2% of men and 52.3% of women). The authors documented that vaccination acceptance was significantly related to the fear of infection or of severe SARS-CoV-2 infection (all *p* < 0.0001).

In the study of Guar et al. [28], the common reasons for vaccine hesitancy were “not yet decided”, fear related to vaccine side effects, and fear of the disease worsening. Concerns about the recurrence of flares and adverse consequences were two of the most common causes of hesitancy in previous studies in the literature [28,33,34]. In our study, fear that the vaccination will aggravate the condition or interfere with the current therapy, concerns about its effectiveness, as well as its side effects were the most prevalent causes for refusing to accept the SARS-CoV-2 vaccine. Other patients were worried about the paucity of the data because individuals with IRD had been omitted from earlier research, resulting in a dearth of information. Such findings should be considered, requiring public education to clear up this uncertainty. Our data showed that the most commonly reported post-vaccination side effects were body ache/pain (32.1%), fatigue (n = 30.3%), and pain at the injection site (29.7%). No serious or life-threatening side effects were self-reported. The study findings highlight that SARS-CoV-2 vaccines have an acceptable safety profile, with the majority of side effects being temporary and mild, which is consistent with previous studies [9,14,35,36]. These results are similar to a cross-sectional analysis of adverse reactions to six SARS-CoV-2 vaccinations in 225 individuals with IRD, which showed that localized pain was the most common adverse event (70.2%). The most prevalent systemic symptoms were fatigue (34.7%), headache (30.6%), and muscular pain (29.3%). No significant adverse events requiring hospitalization or medical treatment were recorded [36]. A recent meta-analysis showed that, except for a higher prevalence of arthralgia, adverse events caused by SARS-CoV-2 vaccinations were comparable in IRD patients to those in healthy controls. The authors concluded that complete immunization is essential in IRD patients. Furthermore, the immunizations were documented to be generally safe [4].

Furer et al. conducted a multicenter trial to determine the immunogenicity and safety of the BNT162b2 mRNA vaccine’s (i.e., the Pfizer-BioNTech vaccine’s) two-dose protocol. Adult IRD patients (n = 686) were compared to a control group (n = 121). Major events that were reported among IRD patients included death (n = 2), non-disseminated herpes zoster (n = 6), uveitis (n = 6), and pericarditis (n = 1). On completing immunization, there was no change in most patients’ disease activity [9]. Nevertheless, a recent study from the European Alliance of Associations for Rheumatology (EULAR) Coronavirus Vaccine (COVAX) physician-reported registry [37] was conducted on participants from 30 countries (n = 5121). Most patients (70%) received the Pfizer/BioNTech vaccine, and 17% received the AstraZeneca/Oxford vaccine. It was encouraging to see that SARS-CoV-2 vaccinations were equally safe in patients with inflammatory/autoimmune rheumatic and musculoskeletal diseases as they were in those with non-inflammatory rheumatic and musculoskeletal diseases [37].

Early results from a web-based survey show that more than 85% of IRD patients did not have a flare after the SARS-CoV-2 vaccine (165 patients, or 14.9%, had flares) [24]. In our study, only nine patients reported disease flares that required medical attention from their rheumatologist. Inflammatory/autoimmune rheumatic and musculoskeletal disease flare-ups and significant adverse events were very infrequent in most individuals who received SARS-CoV-2 vaccination [37]. Raheel et al. reported flares after vaccination, yet they concluded that disease activity was consistent with the disease’s natural history rather than a direct result of SARS-CoV-2 immunizations, which is similar to other published results [38]. Such data is reassuring for IRD patients and physicians when deciding about SARS-CoV-2 vaccination.

This study has a few limitations. First, the lack of random sampling limits the generalizability of these findings to patients with IRD in Kuwait. Second, the study lacks a long-term follow-up for patients, particularly those with disease flares. Nonetheless, the study has several strengths, including adequate sample size and statistical power and the use of a standard questionnaire that has already been developed and used.

## 5. Conclusions

The current study expands the literature by providing novel information on the vaccination rates among IRD patients in Kuwait and by supporting SARS-CoV-2 vaccination safety in this cohort. Future research should explore the long-term outcomes of SARS-CoV-2 vaccination in IRD patients and follow those patients prospectively for possible confounders and predictors of post-vaccination IRD flares and adverse events. Authorities should consider developing educational programs and educational materials to promote understanding and trust in vaccinations and in the potential benefits of vaccination as well as to counter the spread of false information.

## Figures and Tables

**Figure 1 vaccines-11-00666-f001:**
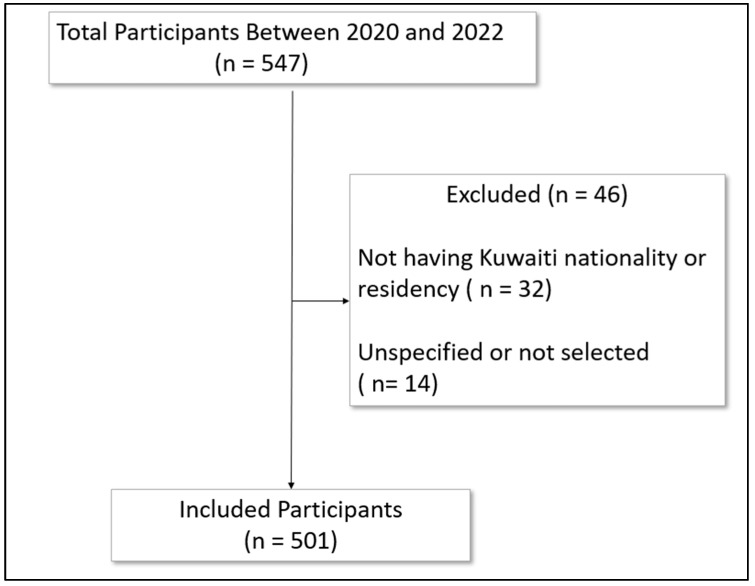
Flow diagram of the included patients.

**Figure 2 vaccines-11-00666-f002:**
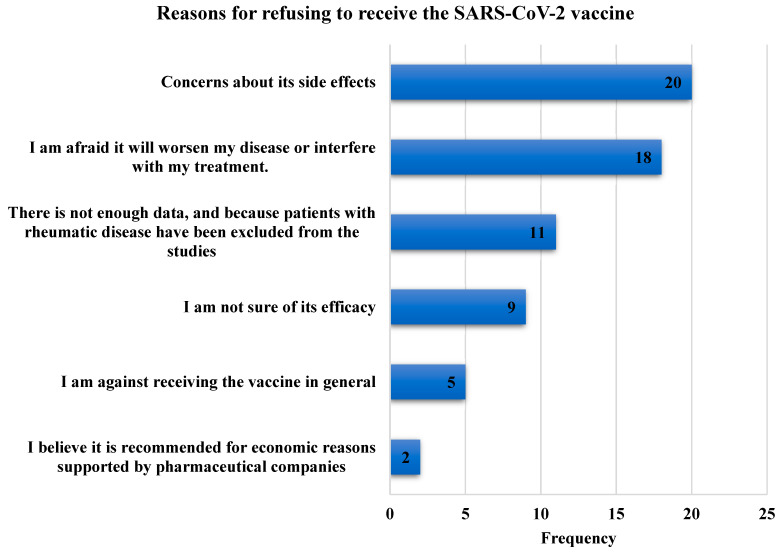
Bar chart showing the frequency of reasons for refusing to receive the SARS-CoV-2 vaccine.

**Figure 3 vaccines-11-00666-f003:**
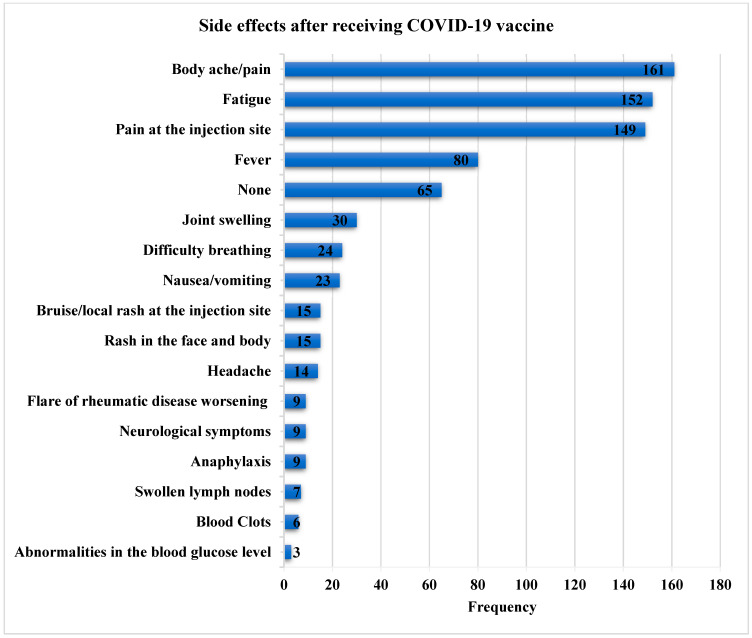
Bar chart showing the frequency of side effects after receiving the SARS-CoV-2 vaccine.

**Table 1 vaccines-11-00666-t001:** Demographic characteristics of the included patients.

		Total = 501
Age, years	Mean ± SD	43.38 ± 11.15
	Range	22.00–78.00
Gender, N (%)	Female	400 (79.8%)
	Male	100 (20.0%)
Educational level, N (%)	≤high school	236 (47.1%)
	>high school	264 (52.7%)
Healthcare workers, N (%)		149(29.7%)
Smoking, N (%)		70 (14.0%)
Rheumatological disease, N (%)	Rheumatoid arthritis	213 (42.5%)
	Spondylarthritis	97 (19.4%)
	Systemic lupus erythematous	95 (19.0%)
	Connective tissue disease	5 (1.0%)
	Rheumatism	1 (0.2%)
	Juvenile arthritis	2 (0.4%)
	Other	88 (17.6%)
Medication, N (%)	Steroid alone	0 (0.00%)
	c DMARD	187 (37.3%)
	b DMARD	90 (18.0%)
	s DMARD	19 (3.8%)
	IVIG	1 (0.2%)
	Anti-TNF	5 (1.0%)
	IL-6	13 (2.6%)
	B-cell depleting therapy	7 (1.4%)
	Combination therapy of DMARD	45 (9.0%)
	Combination of steroid and DMARD therapy	6 (1.2%)
	Combination of DMARD + B-cell depleting therapy	5 (1.0%)
	Other	15 (3.0%)
Disease duration, years	Mean ± SD	10.46 ± 8.54
	Range	1.00–44.00
COVID-19 infection	COVID PCR positive swab result	105 (21.0%)
Patients reported outcomes from the infection, n (%) (How things have turned out?)	I was not hospitalized	86 (17.2%)
	I was hospitalized with supplemental oxygen	6 (1.2%)
	I was hospitalized, but I was not given supplemental oxygen	10 (2.0%)
	I was hospitalized with the ICU admission	1 (0.2%)
Have quarantined during the period from 2019 to 2021, N (%)		49 (9.8%)
Number of vaccinated participants, N (%)		351 (70.1%)
Type of vaccine, N (%)	Pfizer/BioNTech	205 (40.9%)
	AstraZeneca/Oxford	144 (28.7%)

**Table 2 vaccines-11-00666-t002:** Demographic characteristics of the included patients based on vaccination status.

		Vaccinated (n = 351)	Non-Vaccinated (n = 133)	*p*-Value
Age, years	Mean ± SD	43.47 ± 11.53	42.84 ± 10.19	0.621 ^a^
Gender, N (%)	Female	273 (77.8%)	114 (85.7%)	0.052 ^b^
	Male	78 (22.2%)	19 (14.3%)	
Educational level, N (%)	>high school	203 (57.8%)	57 (42.9%)	0.003 ^b,^*
Rheumatological disease, N (%)	Rheumatoid arthritis	154 (43.9%)	53 (39.6%)	0.042 ^b,^*
	Spondylarthritis	69 (19.7%)	28 (20.9%)	
	Systemic lupus erythematous	57 (16.2%)	33 (24.6%)	
	Connective tissue disease	2 (0.6%)	3 (2.2%)	
	Rheumatism	0 (0.0%)	1 (0.7%)	
	Juvenile arthritis	2 (0.6%)	0 (0.0%)	
	Other	67 (19.1%)	16 (11.9%)	
Disease duration, years	Median (Range)	7.00 (1.00–42.00)	7.00 (1.00–44.00)	0.577 ^c^
Patient assessment for the disease, N (describe rheumatic disease)	Inactive	91 (26.0%)	29 (21.8%)	0.059 ^b^
	Mild to moderately active	213 (60.9%)	75 (56.4%)	
	Severely active	46 (13.1%)	29 (21.8%)	
Prior SARS-CoV-2 infection, N (PCR positive swab result)	Yes	73 (20.9%)	32 (24.2%)	0.422 ^b^
	No	277 (79.1%)	100 (75.8%)	
Patient concerns about getting COVID-19, N (%)	Not concerned	4 (10.8%)	7 (41.2%)	0.039 ^b,^*
	Mildly concerned	6 (16.2%)	4 (23.5%)	
	Deeply concerned	15 (40.5%)	3 (17.6%)	
	Moderately concerned	12 (32.4%)	3 (17.6%)	
Prior influenza 2019/2020 combined Prior pneumonia vaccine, N (%)	Yes	152 (43.3%)	32 (23.9%)	<0.001 ^b,^*
	No	199 (56.7%)	102 (76.1%)	
Consulting their doctor, N (%)		288 (82.5%)	76 (57.6%)	<0.001 ^b,^*
Reading KAR online info, N (%)		156 (44.6%)	39 (29.3%)	0.005 ^b,^*
Family member receiving vaccine, N (%)		0 (0.0%)	51 (78.5%)	-

^a^: Independent *t*-test; ^b^: Pearson’s chi-squared test; ^c^: Mann–Whitney test; *: significant at *p* < 0.05.

**Table 3 vaccines-11-00666-t003:** Flares post-vaccination (n = 351).

	Variables	Disease Flare Post-Vaccination (n = 9)	No Flare (n = 342)	*p*-Value
Age	Mean ± SD	42.71 ± 10.44	43.49 ± 11.57	0.861 ^a^
Type of vaccine, N (%)	Pfizer/BioNTech	4 (44.4%)	200 (58.5%)	0.837 ^b^
	AstraZeneca/Oxford	5 (55.6%)	139 (40.6%)	
Gender, N (%)	Female	7 (77.8%)	266 (77.8%)	0.1000 ^b^
	Male	2 (22.2%)	76 (22.2%)	
Smoking, N (%)		1 (11.1%)	54 (15.8%)	0.703 ^b^
Rheumatological disease, N (%)	Rheumatoid arthritis	5 (55.6%)	149 (43.6%)	0.648 ^b^
	Spondylarthritis	0 (0.0%)	69 (20.2%)	
	Systemic lupus erythematous	1 (11.1%)	56 (16.4%)	
	Connective tissue disease	0 (0.0%)	2 (0.6%)	
	Polymyalgia Rheumatica	0 (0.0%)	2 (0.6%)	
	Juvenile arthritis	0 (0.0%)	0 (0.0%)	
	Other	3 (33.3%)	64 (18.7%)	
Patient assessment of the disease, N (%) (Describe rheumatic disease)	Inactive	0 (0.0%)	91 (26.7%)	<0.001 ^b,^*
	Mild to moderately active	4 (44.4%)	209 (61.3%)	
	Severely active	5 (55.6%)	41 (12.0%)	
Patient stopping their medication when taking the vaccine, N (%)		7 (77.8%)	134 (39.2%)	0.141 ^b^

^a^: Independent *t*-test; ^b^: Pearson’s chi-squared test; *: significant at *p* < 0.05.

## Data Availability

Data is contained within the article and survey is available in Appendix A.

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
