# Peer review of "CVAPPS: A Cross-Sectional Study of SARS-CoV-2 Vaccine Acceptance, Perceptions, and Post-Vaccination Side Effects among Rheumatic Disease Patients in Kuwait"

_vaccines, 2023, doi:10.3390/vaccines11030666_

Round 1

Reviewer 1 Report

The authors aimed to estimate the relative numbers of inflammatory rheumatic disease (IRD) patients who have  received the COVID-19 vaccine in the State of Kuwait. Furthermore, this study highlights SARS–CoV-2 vaccine acceptance, perceptions, and post-vaccination side effects among IRD patients. This data should assist healthcare policymakers, public health practitioners, physicians, and rheumatologists in increasing vaccination uptake and awareness in IRD patients.

The study covers some issues that have been overlooked in other similar topics. The structure of the manuscript appears adequate and well divided in the sections. Moreover, the study is easy to follow, but some issues should be improved. Some of the comments that would improve the overall quality of the study are:

I-) Authors must pay attention to the technical terms acronyms they used in the text

II-) Conclusion Section: This paragraph required a general revision to eliminate redundant sentences and to add some "take-home message".

Author Response

Response to Reviewer 1 Comments

Point 1: The authors aimed to estimate the relative numbers of inflammatory rheumatic disease (IRD) patients who have received the COVID-19 vaccine in the State of Kuwait. Furthermore, this study highlights SARS–CoV-2 vaccine acceptance, perceptions, and post-vaccination side effects among IRD patients. This data should assist healthcare policymakers, public health practitioners, physicians, and rheumatologists in increasing vaccination uptake and awareness in IRD patients. The study covers some issues that have been overlooked in other similar topics. The structure of the manuscript appears adequate and well divided in the sections. Moreover, the study is easy to follow, but some issues should be improved.

Response 1: Thank you very much.

Point 2: Authors must pay attention to the technical terms acronyms they used in the text.

Response 2: Thank you for pointing this out to us. We have considered this in the modified version of the manuscript.

Point 3: Conclusion Section: This paragraph required a general revision to eliminate redundant sentences and to add some "take-home message"

Response 3: It is now modified as requested (highlighted on page 10 in the modified version).

Reviewer 2 Report

There are some revisions that I would like the authors to address and/or to consider

Abstract: Methodology:  How COVID-19 Vaccine Acceptance and Perceptions were measured. Present used Statistical Software?

Main manuscript: Methods:  present used instruments specially for Vaccine Acceptance and Perceptions.  How “Reasons for refusing to receive the COVID-19 vaccine” were assessed?

Study size:  (50% in an undetermined population size with 5% precision, the minimum sample size 90 required for this study is 398 patients):  , it seems that a census method has been used .

Author Response

Response to Reviewer 2 Comments

Point 1: There are some revisions that I would like the authors to address and/or to consider.

Abstract: Methodology: How COVID-19 Vaccine Acceptance and Perceptions were measured. Present used Statistical Software?

Response 1: Thank you for the time and input. We highlighted in the methods that the self-administrated questionnaire was used to collect the relevant data and added more details in the methods section, and the statistical software is added as requested (highlighted on page 1 in the modified version).

Point 2: Main manuscript: Methods: present used instruments specially for Vaccine Acceptance and Perceptions. How “Reasons for refusing to receive the COVID-19 vaccine” were assessed?

Response 2: Thanks for pointing this out to us; we have described it (highlighted on page 2 in the modified version). The used instrument is attached as a Supplementary file. The reasons were captured from the answers received from the responses to section 6 in the questionnaire (Opposition to receiving the vaccine).

Point 3: Study size: (50% in an undetermined population size with 5% precision, the minimum sample size 90 required for this study is 398 patients):  , it seems that a census method has been used .

Response 3: We calculated the sample size based on the recommended methods for cross-sectional studies as reported in a previous sample size calculation guide (PMID: 31172113), considering that the primary endpoint of this study is to determine the rates of vaccination status and post-vaccination side effects and flares among the accessible population (IRD patients in the State of Kuwait). However, the sample size calculation resulted in 398 patients; we managed to include 501 IRD patients from 7 governmental hospitals in Kuwait.

Reviewer 3 Report

The article entitled “CVAPPS: A Cross-sectional Study of COVID-19 Vaccine Acceptance, Perceptions, and Post-vaccination Side Effects Among  Rheumatic Disease Patients in Kuwait” has been reviewed.

This is a well written cross-sectional study to assess SARS CoV2 vaccine uptake and acceptance by patients with IRD in the State of Kuwait. Authors have taken into account STROBE criteria for the quality dissemination f observational study results. The topic on which the authors focus is truly of great relevance worldwide.

In all I believe this paper has the desired quality to be published as is.

Just one comment  on the naming of the vaccine. Vaccine is targeted to prevent disease caused by a certain virus  for example: Human Papiloma virus vaccine , and in this line I would prefer naming the vaccine against COVID-19 as SARS-CoV-2 vaccine. But I understand that this is more a sematic issue rather than for the clarity of the paper. Therefore I leave the choice to change to the editor 

Author Response

Response to Reviewer 3 Comments

Point 1: This is a well written cross-sectional study to assess SARS CoV2 vaccine uptake and acceptance by patients with IRD in the State of Kuwait. Authors have taken into account STROBE criteria for the quality dissemination f observational study results. The topic on which the authors focus is truly of great relevance worldwide. In all I believe this paper has the desired quality to be published as is.

Response 1: Thank you very much.

Point 2: Just one comment on the naming of the vaccine. Vaccine is targeted to prevent disease caused by a certain virus for example:  Human Papiloma virus vaccine, and in this line I would prefer naming the vaccine against COVID-19 as SARS-CoV-2 vaccine. But I understand that this is more a sematic issue rather than for the clarity of the paper. Therefore I leave the choice to change to the editor

Response 2: Thank you for pointing this out to us. We have modified it throughout the manuscript.

Round 2

Reviewer 3 Report

The authors have made all suggested changes appropriately